# Heterodimerization of UNC-13/RIM regulates synaptic vesicle release probability but not priming in *C. elegans*

**Haowen Liu[1†], Lei Li[1†], Daniel Nedelcu[2,3‡], Qi Hall[2,3‡], Lijun Zhou[4], Wei Wang[1], Yi Yu[1], Joshua M Kaplan[3]\*, Zhitao Hu[1]\***

[1]Clem Jones Centre for Ageing Dementia Research, Queensland Brain Institute, The University of Queensland, Brisbane, Australia; [2]Department of Molecular Biology, Massachusetts General Hospital, Boston, United States; [3]Department of Neurobiology, Harvard Medical School, Boston, United States; [4]Center for Computational and Integrative Biology, Massachusetts General Hospital, Howard Hughes Medical Institute, Boston, United States

**Abstract** UNC-13 proteins play an essential role in synaptic transmission by recruiting synaptic vesicles (SVs) to become available for release, which is termed SV priming. Here we show that the C2A domain of UNC-13L, like the corresponding domain in mammalian Munc13-1, displays two conserved binding modes: forming C2A/C2A homodimers, or forming a heterodimer with the zinc finger domain of UNC-10/RIM (C2A/RIM). Functional analysis revealed that UNC-13L's C2A promotes synaptic transmission by regulating a post-priming process. Stimulus-evoked release but not SV priming, was impaired in *unc-10* mutants deficient for C2A/RIM heterodimerization, leading to decreased release probability. Disrupting C2A/C2A homodimerization in UNC-13L-rescued animals had no effect on synaptic transmission, but fully restored the evoked release and the release probability of *unc-10*/RIM mutants deficient for C2A/RIM heterodimerization. Thus, our results support the model that RIM binding C2A releases UNC-13L from an autoinhibitory homodimeric complex to become fusion-competent by functioning as a switch only.
DOI: https://doi.org/10.7554/eLife.40585.001

**\*For correspondence:**
kaplan@molbio.mgh.harvard.edu (JMK);
z.hu1@uq.edu.au (ZH)

[†]These authors contributed equally to this work
[‡]These authors also contributed equally to this work

## Introduction

Neurotransmitters are released by $Ca^{2+}$-triggered synaptic vesicle (SV) exocytosis, a processmediated by the SNARE complex, and regulated by synaptic proteins such as Munc18s, Munc13s, RIMs, and synaptotagmins (*Südhof and Rizo, 2011*; *Jahn and Fasshauer, 2012*). Before final fusion with the plasma membrane, SVs undergo two crucial steps: docking and priming. These processes ensure that the SVs are translocated to the plasma membrane and become fusion-competent at the active zone (*Sudhof, 2004*). Defects in docking or priming produce corresponding decreases in SV exocytosis (*Aravamudan et al., 1999*; *Augustin et al., 1999*; *Richmond et al., 1999*; *Hammarlund et al., 2007*; *Imig et al., 2014*). However, post-priming regulation of SV release has also been reported (*Pang et al., 2006*; *Bacaj et al., 2015*; *Luo et al., 2015*). Various studies have demonstrated that Munc13-1, a protein with multiple functional domains, is required for the docking and priming (*Aravamudan et al., 1999*; *Augustin et al., 1999*; *Richmond et al., 1999*), as well as post-priming $Ca^{2+}$-triggered SV release (*Basu et al., 2007*; *Chen et al., 2013*; *Hu et al., 2013*). The functions of Munc13-1 in these processes are believed to be mediated by its central MUN domain, which directly interacts with syntaxin and promotes SNARE assembly (*Ma et al., 2011*; *Yang et al., 2015*).

Multiple domains of Munc13-1 have been implicated in regulating the MUN domain's SV priming activity (*Michelassi et al., 2017*; *Xu et al., 2017*). In mammals, four Munc13 isoforms have been

identified, two of which contain an N-terminal C2A domain (Munc13-1 and ubMunc13-2) (*Brose et al., 1995*; *Song et al., 1998*; *Betz et al., 2001*). Betz et al. first found that the C2A domain binds to the zinc finger (ZF) domain of RIM, an active zone protein that is required for SV priming (*Betz et al., 2001*). A double mutation in rat RIM2α's ZF domain (K97/99E) disrupts C2A/RIM heterodimerization (*Dulubova et al., 2005*) and was subsequently shown to produce a large reduction in the readily releasable pool (RRP) of SVs (*Deng et al., 2011*; *Camacho et al., 2017*), indicating that C2A binding to RIM plays an important role in SV priming. Apart from forming a heterodimer with RIM, the C2A domain also forms a homodimer, thereby promoting formation of Munc13-1 homodimers (*Lu et al., 2006*). This homodimerization, however, is not necessary for SV priming and postpriming SV release (*Camacho et al., 2017*).

In contrast, studies in chromaffin cells have shown that a truncated Munc13-1 lacking the N-terminal C2A domain is fully able to mediate vesicle priming (*Betz et al., 2001*), suggesting that the role of Munc13-1 in priming is independent of the C2A domain and its binding with RIM. Similarly, a recent study in *C. elegans* found that removing the N-terminal C2A domain from UNC-13L, a Munc13-1 homolog, does not cause a change in the size of the RRP (*Zhou et al., 2013a*). These conflicting findings have called into doubt whether binding of the C2A domain with RIM is indeed playing a role in SV priming, prompting us to re-analyze the function of both C2A/RIM heterodimerization and C2A/C2A homodimerization using an in vivo system.

Here, we examined the functional importance of the C2A domain by generating a transgenic line of UNC-13L with a point mutation in the C2A domain (K49E) that blocks formation of C2A/C2A homodimers, and a mutant carrying two mutations in the *unc-10*/RIM ZF domain (K77/79E) that disrupts C2A/RIM heterodimer formation. We found that tonic neurotransmitter release at a low $Ca^{2+}$ concentration and stimulus-evoked release were significantly reduced in the *unc-10*(*nu487* K77/79E) mutants, whereas SV release was normal in the UNC-13L(K49E)-rescued *unc-13* mutants. Moreover, the hypertonic sucrose-evoked responses were unaltered following disruption of homodimerization and heterodimerization, suggesting that SV priming is not determined by the state of the C2A domain. Interestingly, simultaneously disrupting C2A/C2A homodimerization in the *unc-10*(*nu487* K77/79E) mutant fully restored both tonic and evoked release. These results suggest that the synaptic transmission defect in the *unc-10*(*nu487* K77/79E) mutant is most likely caused by formation of more C2A/C2A homodimers, and that the role of UNC-10/RIM binding UNC-13 C2A domain is to release the C2A from an autoinhibitory state, allowing it to become a fusion-competent monomeric form. Finally, we show that UNC-13's C2A domain increases release probability but has little effect on SV priming.

## Results

### C2A/C2A homodimerization and C2A/RIM heterodimerization are conserved in *C. elegans*

The UNC-13L C2A domain and the UNC-10/RIM ZF domain share high sequence identity with the corresponding domains of mouse Munc13-1 (50%) and RIM (48%) (*Figure 1A,B*), suggesting that UNC-13's C2A would form homo- and heterodimers in a manner similar to mouse Munc13-1. To test this idea, we directly assessed homo- and heterodimer formation by the UNC-13 C2A domain. Gel filtration analysis indicated that recombinant C2A forms a homodimer, and that this homodimerization was disrupted by a mutation (K49E) analogous to that which disrupts Munc13-1 homodimerization (*Figure 1A,C*) (*Lu et al., 2006*; *Deng et al., 2011*). To determine whether UNC-13L forms a heterodimer with UNC-10/RIM, we performed isothermal titration calorimetry (ITC) and co-immunoprecipitation experiments. Our results showed that the UNC-13L C2A domain tightly binds the UNC-10/RIM ZF domain (Kd = 1.29 μM, *Figure 1D,E*), and that this interaction was eliminated by a double mutation in the ZF domain (K77/79E), which is analogous to mutations that disrupt binding of the corresponding mammalian Munc13-1 and RIM domains (*Figure 1D–F*) (*Dulubova et al., 2005*). Thus, our findings confirmed that UNC-13L C2A has the same two binding modes exhibited by mouse Munc13-1.

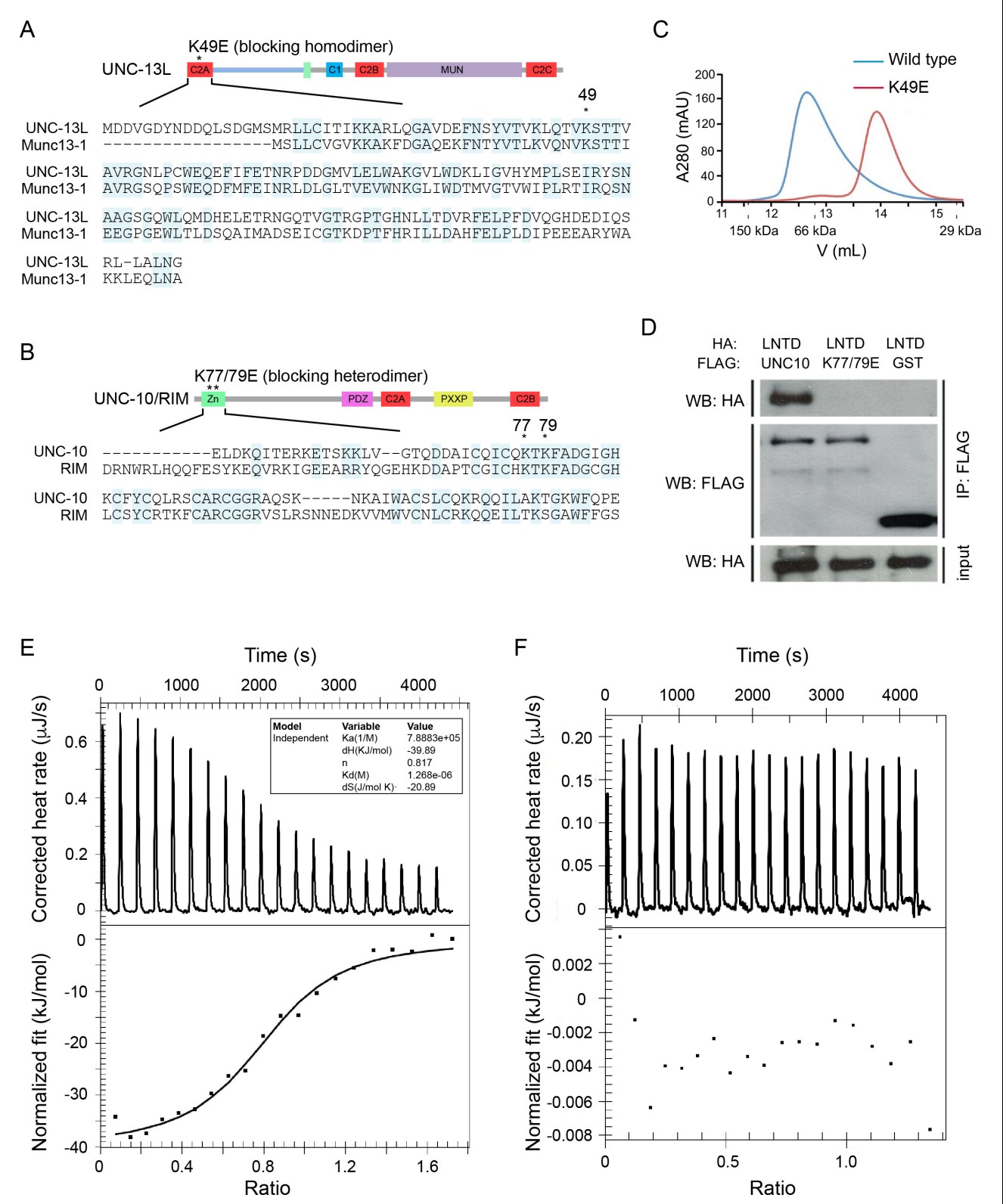

**Figure 1.** UNC-13L exhibits conserved homodimerization and heterodimerization in *C. elegans*. (A, B) Sequence alignment of the C2A domain between worm UNC-13L and rat Munc13-1, and the zinc finger (ZF) domain between worm UNC-10 and mouse RIM1. The conserved residues in the C2A domain (K49) and the ZF domain (K77, K79) that disrupt homodimer and heterodimer formation are indicated by stars. (C) Separation of purified UNC-13L C2A monomer and homodimer by gel filtration. The wild-type C2A forms a homodimer that runs at 12.57 mL (blue trace, corresponding to an apparent

*Figure 1 continued on next page*

*Figure 1 continued*
weight of 89 kDa); this homodimer is disrupted in the monomeric UNC-13L(K49E) mutant which runs at 13.88 mL (apparent weight 52 kDa). Elution volumes of gel filtration standards are indicated underneath the plot, as follows: 11.62 mL for alcohol dehydrogenase (150 kDa), 12.85 mL for bovine serum albumin (66 kDa), and 15.45 mL for carbonic anhydrase (29 kDa), with the column void volume being 10.78 mL. (D) Co-immunoprecipitation (co-IP) shows that HA-LNTD (UNC-13L N-terminal domain, aa 1″605) binds wild-type FLAG-UNC-10 (aa 1–601) but not UNC-10(K77/79E) or GST. These results are representative of three independent experiments. (E, F) Isothermal titration calorimetry (ITC) analysis of binding of wild-type or mutated (K77/79E) UNC-10 with UNC-13L.
DOI: https://doi.org/10.7554/eLife.40585.002

## Disrupting C2A/RIM heterodimerization reduces mEPSCs/mIPSCs in low $Ca^{2+}$

We next examined the functional roles of the C2A/C2A homodimerization and C2A/RIM heterodimerization in tonic neurotransmitter release by measuring miniature excitatory/inhibitory postsynaptic currents (mEPSCs/mIPSCs) at the neuromuscular junction. Both the mEPSCs and the mIPSCs were recorded in the presence of 0 mM and 1 mM $Ca^{2+}$, as we have previously found that the mIPSCs are almost arrested in 0 mM $Ca^{2+}$ but are normal 1 mM $Ca^{2+}$ in some mutants (*Liu et al., 2018*), suggesting that different $Ca^{2+}$ concentrations should be tested to provide an accurate and thorough evaluation of tonic release. Our results showed that the mEPSCs and mIPSCs were completely abolished in the *unc-13(s69)* null mutant, and were fully restored by neuronal expression of full-length UNC-13L (*Figure 2*). Deleting the C2A domain (UNC-13LΔC2A) significantly decreased mEPSC frequency in both 0 mM and 1 mM $Ca^{2+}$ (*Figure 2B,E*, *Table 1*), demonstrating an important role of the C2A domain in tonic acetylcholine (ACh) release. The mIPSC frequency was significantly decreased in the UNC-13LΔC2A rescue animals in 0 mM $Ca^{2+}$ (*Figure 2C*, *Table 1*), but was unchanged in 1 mM $Ca^{2+}$, compared to that in the UNC-13L rescue animals (*Figure 2F*). Thus, the UNC-13L C2A domain is important for tonic release of both ACh and GABA; however, the requirement for C2A in GABA release can be bypassed by the presence of extracellular calcium, presumably because of increased release probability. The differential impact of deleting C2A on ACh and GABA release in 1 mM $Ca^{2+}$ could reflect differences in release probability at these two types of synapses (*Liu et al., 2018*). In fact, SV release from inhibitory synapses displays a higher release probability than excitatory synapses in the mammalian central nervous system (*Zhou et al., 2013b*; *Zhou et al., 2013c*). Deleting UNC-13L's C2A domain could also alter other aspects of synapse structure, such as SV trafficking, and coupling of SVs to $Ca^{2+}$ entry at the release sites.

We next asked whether C2A/C2A homodimerization or C2A/RIM heterodimerization accounts for the role of the C2A domain in tonic release. To investigate this, we first examined the mEPSCs and mIPSCs in the *unc-13* mutants rescued by UNC-13L carrying the K49E mutation, which blocks C2A/C2A homodimerization (*Figure 1B*). The mEPSC and mIPSC frequencies in the UNC-13L(K49E) rescue animals were comparable to those in the UNC-13L rescue animals in both 0 mM and 1 mM $Ca^{2+}$ conditions (*Figure 2A–F*, *Table 1*), indicating that homodimerization does not affect tonic ACh or GABA release. To determine the role of C2A/RIM heterodimerization, we used crispr to introduce a K77/79E double mutation in the UNC-10 ZF domain, *unc-10(nu487 K77/79E)*. Compared to wild-type animals, *unc-10(nu487 K77/79E)* mutants display slightly uncoordinated behaviour (data not shown). Analysis of tonic release showed that the mEPSCs and mIPSCs were markedly reduced in *unc-10(nu487 K77/79E)* mutants in 0 mM $Ca^{2+}$, whereas both were unaltered in 1 mM $Ca^{2+}$ (*Figure 2A–F*), suggesting that C2A/RIM heterodimerization is only required for tonic release in low $Ca^{2+}$ when release probability is reduced. To confirm this, we measured mEPSCs and mIPSCs in two additional $Ca^{2+}$ concentrations (0.05 mM and 0.1 mM). As shown in *Figure 2G*, increasing the $Ca^{2+}$ concentration from 0 mM to 0.05 mM and 0.1 mM caused robust enhancement of the mEPSC and mIPSC frequencies in wild-type animals. In *unc-10(nu487 K77/79E)* mutants, a significantly lower mIPSC frequency was observed in 0.05 mM $Ca^{2+}$ but not in 0.1 mM $Ca^{2+}$, whereas mEPSCs were unaltered in both 0.05 mM and 0.1 mM $Ca^{2+}$ (*Figure 2G*). Our results therefore demonstrate that C2A/RIM heterodimerization promotes tonic release but that this effect is bypassed when release probability is increased by raising external $Ca^{2+}$ concentrations.

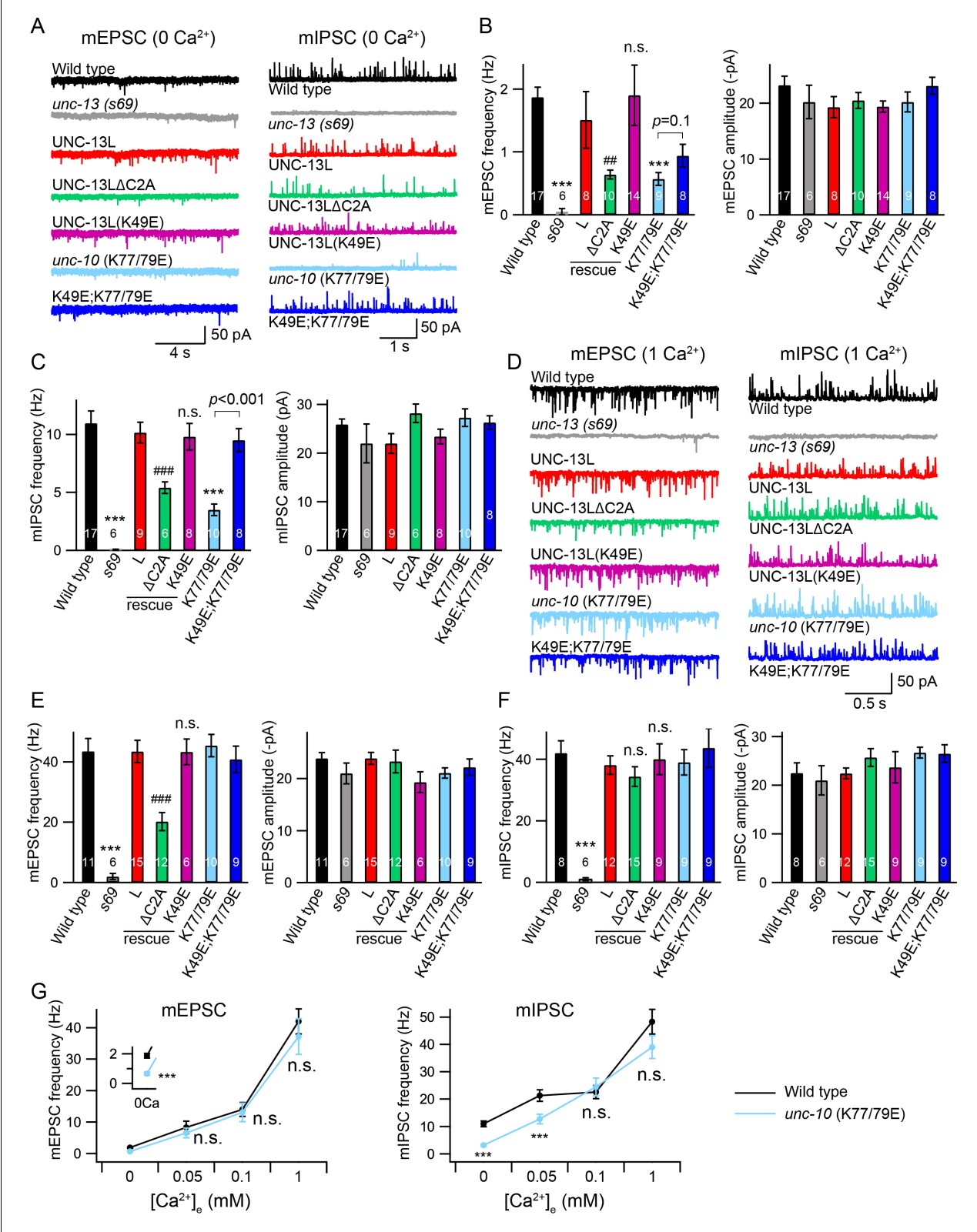

**Figure 2.** Tonic release is reduced by disrupting C2A/RIM heterodimerization in low $Ca^{2+}$. (**A**) Representative traces of mEPSCs and mIPSCs recorded at 0 mM $Ca^{2+}$ from indicated genotypes. (**B, C**) Averaged frequency and amplitude of the mEPSCs and mIPSCs. (**D**) Representative traces of mEPSCs and mIPSCs recorded at 1 mM $Ca^{2+}$ from the same genotypes in (**A**). (**E, F**) Quantification of the mEPSC and mIPSC frequency and amplitude from the genotypes in (**D**). Data are mean ± SEM (##, p < 0.01, ###, p < 0.001, n.s., non-significant when compared to UNC-13L rescue; ***, p < 0.001 when

*Figure 2 continued on next page*

*Figure 2 continued*

compared to wild-type; one-way ANOVA). The number of worms analyzed for each genotype is indicated in the bar graphs. (G) Quantification of the mEPSC and mIPSC frequencies at various $Ca^{2+}$ levels (0, 0.05, 0.1, and 1 mM) from wild-type (black) and unc-10(nu487 K77/79E) mutants (red). Data are mean ± SEM (***, p < 0.001, n.s., non-significant when compared to wild-type; student's t-test).
DOI: https://doi.org/10.7554/eLife.40585.003

## A monomeric C2A domain is fully functional for tonic release

The decreased mEPSC frequency observed in the UNC-13LΔC2A rescue animals at 1 mM $Ca^{2+}$ indicates that the C2A domain may have an additional function in regulating tonic release, particularly at high $Ca^{2+}$. We therefore examined whether simultaneously blocking the homo- and heterodimerization of the C2A domain could account for this additional function. To do that, we made UNC-13L (K49E); unc-10(nu487 K77/79E) double mutant by crossing the unc-10(nu487 K77/79E) into the UNC-13L(K49E) rescue strain. In 0 mM $Ca^{2+}$ recordings, the UNC-13L(K49E); unc-10(nu487 K77/79E) double mutant mIPSC rate was significantly higher than that observed in unc-10(nu487 K77/79E) single mutants, suggesting that blocking homodimer formation was sufficient to prevent the decreased tonic GABA release caused by the unc-10(nu487 K77/79E) mutation (*Figure 2C*). By contrast, blocking homodimerization did not significantly improve mEPSC rates in 0 mM $Ca^{2+}$ in unc-10(nu487 K77/79E) mutants (*Figure 2B*). Thus, heterodimerization increases tonic GABA release by preventing homodimer formation, whereas heterodimerization promotes tonic ACh release by a distinct mechanism. In addition, the tonic ACh and GABA release were at wild-type level in the UNC-13L(K49E); unc-10(nu487 K77/79E) double mutant in 1 mM $Ca^{2+}$, similar to those in the UNC-13L(K49E) rescue and the unc-10(nu487 K77/79E) mutant (*Figure 2E,F*). Taken together, these results demonstrate that blocking C2A homo- and heterodimerization does not decrease tonic ACh and GABA release. These results differ from those obtained in the UNC-13LΔC2A rescue animals, suggesting that the

**Table 1.** Comparison of SV release.

| | Tonic release | | | | | | | | Evoked EPSC | | RRP | $P_{vr}$ |
|---|---|---|---|---|---|---|---|---|---|---|---|---|
| | mEPSC (0 mM $Ca^{2+}$) | | mEPSC (1 mM $Ca^{2+}$) | | mIPSC (0 mM $Ca^{2+}$) | | mIPSC (1 mM $Ca^{2+}$) | | | | | |
| | Frequency (Hz) | Amplitude (-pA) | Frequency (Hz) | Amplitude (-pA) | mIPSC (Hz) | Amplitude (pA) | mIPSC (Hz) | Amplitude (pA) | Amplitude (-nA) | Charge (-pC) | Charge (-pC) | Charge (-pC) |
| Wild-type | 1.87 ± 0.16 | 23.2 ± 1.6 | 43.5 ± 4.3 | 23.9 ± 1.1 | 11 ± 1.1 | 25.9 ± 1.1 | 42.5 ± 4.1 | 22.5 ± 2.1 | 2.27 ± 02. | 20.6 ± 2.1 | 221 ± 23.1 | 0.12 ± 0.01 |
| unc-13 (s69) | 0.05 ± 0.05 | 20.2 ± 3 | 2 ± 1 | 21.1 ± 2.1 | 0.05 ± 0.05 | 22.1 ± 4.1 | 1.1 ± 0.5 | 21.5 ± 3.2 | 0.02 ± 0.01 | 0.05 ± 0.01 | 25.5 ± 7.1 | 0 |
| UNC-13L | 1.51 ± 0.45 | 19.3 ± 1.9 | 43.4 ± 3.7 | 23.9 ± 1.1 | 10.1 ± 0.9 | 22.3 ± 2.2 | 38.1 ± 3.5 | 22.4 ± 1.1 | 1.7 ± 0.14 | 15.4 ± 2.6 | 237 ± 25.4 | 0.06 ± 0.01 |
| UNC-14LΔC2A | 0.64 ± 0.07 | 20.5 ± 1.4 | 20.2 ± 3 | 23.3 ± 2.2 | 5.4 ± 0.5 v | 28.2 ± 1.9 | 34.4 ± 3.2 | 25.7 ± 1.8 | 1.08 ± 0.12 | 8.4 ± 1.6 | 248 ± 66.3 | 0.034 ± 0.006 |
| UNC-13L (K49E) | 1.9 ± 0.48 | 19.4 ± 1.1 | 43.3 ± 4.3 | 19.3 ± 2.4 | 9.8 ± 1.2 | 23.4 ± 1.5 | 40.6 ± 5.1 | 23.7 ± 3.2 | 2.1 ± 0.27 | 17.3 ± 2.3 | 243 ± 38.3 | 0.07 ± 0.01 |
| unc-10 (K77/79E) | 0.57 ± 0.1 | 20.2 ± 1.8 | 45.4 ± 3.7 | 21.1 ± 1.2 | 3.5 ± 0.5 | 27.3 ± 1.8 | 39.7 ± 4.1 | 26.7 ± 1.1 | 1.18 ± 0.11 | 11.5 ± 1.5 | 245 ± 28.2 | 0.046 ± 0.006 |
| K49E; K77/79E | 0.94 ± 0.18 | 23.1 ± 1.5 | 40.8 ± 4.4 | 22.2 ± 1.6 | 9.5 ± 1 | 26.3 ± 1.4 | 43.7 ± 6.3 | 26.5 ± 1.8 | 1.81 ± 0.15 | 18.1 ± 2.2 | 260 ± 40 | 0.07 ± 0.008 |

All data were collected from three independent experiments to ensure the observed phenotype are replicable.
DOI: https://doi.org/10.7554/eLife.40585.004

effect of deleting the C2A domain on the tonic release is not a result of a lack of homo- and heterodimerization.

## Stimulus-evoked release is reduced when C2A/RIM heterodimerization is blocked

Next, we analysed the impact of C2A interactions on stimulus-evoked ACh release. Compared to UNC-13L, UNC-13LΔC2A exhibited a smaller evoked EPSC (*Figure 3A*, *Table 1*), consistent with previous findings (*Hu et al., 2013*; *Zhou et al., 2013a*). The evoked EPSCs were significantly smaller in *unc-10*(*nu487* K77/79E) mutants, exhibiting decreased amplitude and total charge transfer compared to wild-type animals (*Figure 3A–C*). The percentage decrease of the evoked charge transfer in the *unc-10*(*nu487* K77/79E) mutant (44%) was similar to that observed in the UNC-13LΔC2A rescue animals (46%), suggesting that C2A/RIM heterodimerization accounts for the role of the C2A domain in evoked release. Thus, our results revealed that C2A/RIM heterodimerization is required for evoked neurotransmitter release. It should be noted that the evoked EPSCs were recorded in 1 mM Ca$^{2+}$ solution in which, however, the mEPSCs were unchanged (*Figure 2E*). These different observations on tonic and evoked neurotransmitter release may reflect differential release probabilities of the two release modes at various Ca$^{2+}$ levels. Indeed, the mEPSCs and the stimulus-evoked EPSCs exhibit large differences in their Ca$^{2+}$ dependence (*Williams et al., 2012*).

Next, we asked whether C2A homodimers regulate evoked responses. The evoked EPSCs mediated by UNC-13L(K49E) were indistinguishable from those rescued by wild-type UNC-13L, indicating that C2A homodimers are not required for evoked responses (*Figure 3A–C*). To determine whether homodimers inhibited evoked ACh release, we analyzed UNC-13L(K49E); *unc-10* (*nu487* K77/79E) double mutants. We found that preventing C2A/C2A homodimerization in the *unc-10*(*nu487* K77/79E) mutant fully restored the evoked EPSC to a wild-type level (*Figure 3A–C*). These results suggest that the reduced evoked EPSC caused by blocking C2A/RIM heterodimerization is mediated by increased formation of C2A homodimers. These results are consistent with prior analysis of mouse synapses, which suggest that Munc13-1 homodimers inhibit SV release (*Deng et al., 2011*). These results also demonstrate that the monomeric C2A domain is sufficient to support evoked release, consistent with the observation of tonic release (*Figure 2*, *Table 1*). Thus, the RIM ZF domain appears to function only as a switch that releases the C2A domain from the inhibitory homomeric complex.

## The C2A domain is not required for SV priming

A decrease in evoked EPSCs may arise from a priming defect or a change in release probability. We therefore examined whether SV priming is regulated by the C2A domain and its two different states (homodimer and heterodimer). SV priming was measured by the pulsed application of hypertonic sucrose solution (1M, 2 s) onto the ventral nerve cord, and the size of the RRP was calculated by integrating the sucrose-evoked current. The current was almost eliminated in the *unc-13* null mutants, and was fully restored by expressing a wild-type UNC-13L transgene in all neurons (*Figure 3E*). Strikingly, UNC-13L lacking the C2A domain could still restore the sucrose-evoked current to a wild-type level (*Figure 3E,F*, *Table 1*), indicating that the C2A domain is not required for SV priming. These results differ from those reported in mouse hippocampal neurons, in which the RRP was largely reduced when the Munc13-1 C2A domain was removed (*Camacho et al., 2017*). Despite these differences, our results, together with those obtained in chromaffin cells in which Munc13-1 lacking the C2A domain enhances secretion to as high a level as does Munc13-1 (*Betz et al., 2001*), support the notion that truncated UNC-13 lacking the C2A domain is still able to prime SVs.

The C2A deletion had no effect on SV priming, suggesting that C2A/C2A homodimerization and C2A/RIM heterodimerization are also not required for priming. Nevertheless, we assessed the RRP size in the UNC-13L(K49E) rescue animals and the *unc-10*(K77/79E) mutant by puffing hypertonic sucrose (1M, 2 s). Our data showed that the sucrose-evoked currents in these animals were comparable to those observed in UNC-13L rescue and wild-type animals (*Figure 3E,F*), demonstrating that homodimerization or heterodimerization is not playing a role in SV priming. The unchanged SV priming in the *unc-10*(K77/79E) mutant was also confirmed by puffing a lower concentration of sucrose (0.5M, 5 s; data not shown). Consequently, the probability of synaptic vesicle release (P$_{vr}$) calculated by the ratio of the charge transfer of the evoked EPSCs to the charge transfer of the sucrose-evoked

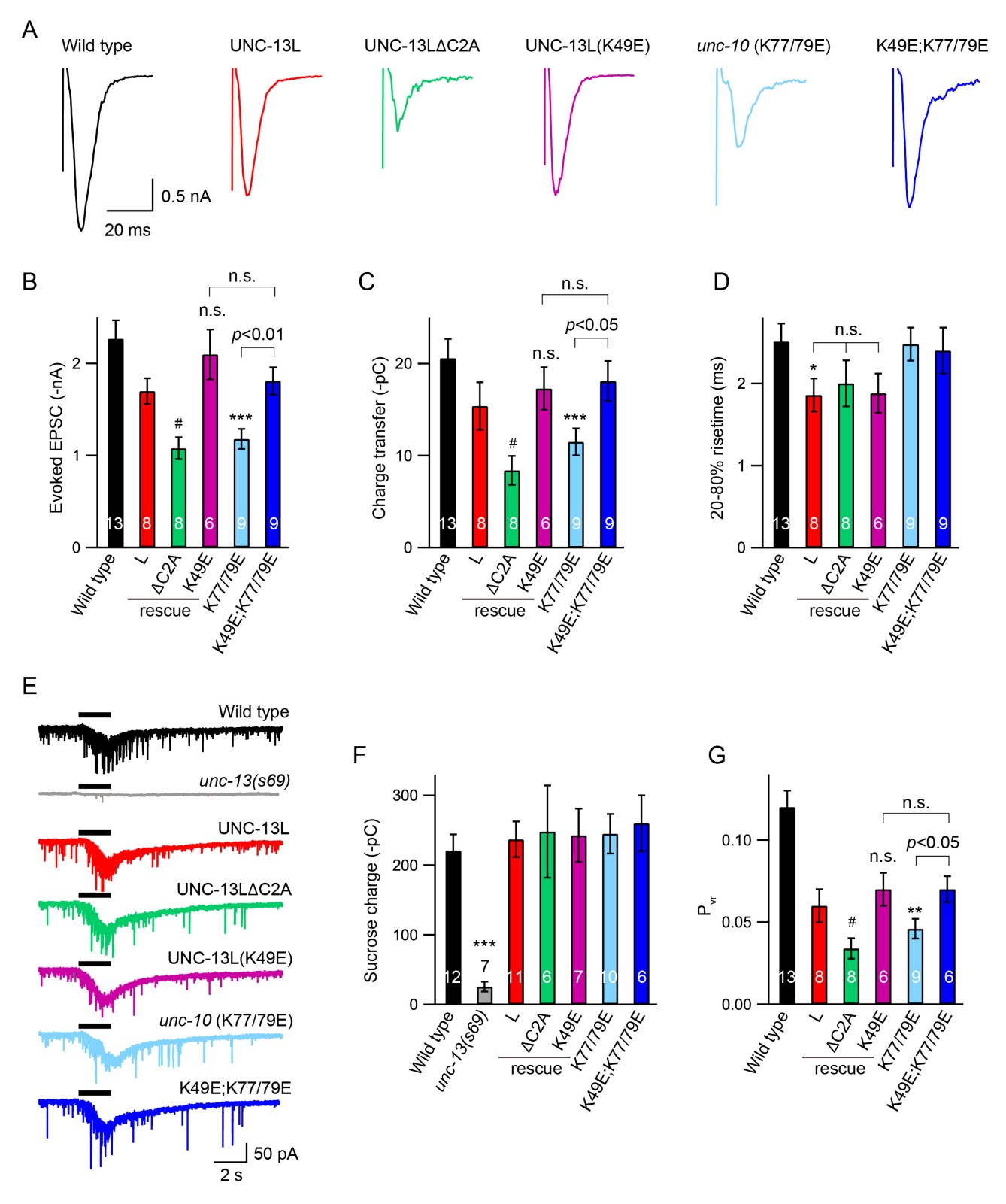

**Figure 3.** Effects of homodimerization and heterodimerization on evoked neurotransmitter release. Electronic stimulus and hypertonic sucrose-evoked EPSCs were recorded from the body wall muscle of adult animals in 1 mM Ca$^{2+}$. (**A**) Example traces of the evoked EPSCs from indicated genotypes. (**B–D**) Quantification of the amplitude, charge transfer and 20–80% risetime of the evoked EPSCs in (**A**). Example traces and averaged charge transfer of the sucrose-evoked EPSCs from the same genotypes in (**A**) are shown in (**E**) and (**F**). (**G**) Quantification of the probability of SV release (P$_{vr}$). Data are

*Figure 3 continued on next page*

Figure 3 continued

mean ± SEM (*, p < 0.05, **, p < 0.01, ***, p < 0.001 when compared to wild-type; #, p < 0.05, n.s., non-significant when compared to UNC-13L rescue; one-way ANOVA).

DOI: https://doi.org/10.7554/eLife.40585.005

The following figure supplement is available for figure 3:

**Figure supplement 1.** Quantification of the leak currents from sucrose responses.

DOI: https://doi.org/10.7554/eLife.40585.006

currents was significantly reduced in the ΔC2A rescue and the *unc-10*(K77/79E) mutant (*Figure 3G*). It should be noted that recording of the sucrose-evoked current requires a tight gigaohm seal between the recording pipette and the cell, as a loose seal often leads to an unreal current. We therefore analyzed the leak currents for each sucrose recording. Our data showed that all recordings were controlled well with acceptable leak currents (*Figure 3—figure supplement 1*). Together, our results demonstrate that the UNC-13L C2A domain regulates release probability by interacting with the UNC-10/RIM, rather than by regulating SV priming.

The decreased release probability in the UNC-13LΔC2A rescue and the *unc-10*(K77/79E) mutant could arise from a change in the expression level of UNC-13, or a change in sub-localization of the UNC-13 proteins at the nerve terminals. We next analyzed the UNC-13 protein levels from different transgenic rescue lines. A fluorescence protein (mApple) was fused to the C terminus of the various UNC-13L constructs (WT, K49E, and ΔC2A), and expressed in *unc-13* null mutants. Our prior study showed that c-terminal mCherry tags do not interfere with UNC-13's function in promoting SV exocytosis (*Hu et al., 2013*). All mApple-tagged UNC-13L constructs rescued the SV release in the *unc-13* mutants, and the phenotypes were similar to those rescued by corresponding untagged UNC13L constructs (*Figure 4—figure supplement 1*). By quantifying the mApple fluorescence intensity from different rescue lines, we showed that there were no significant differences (*Figure 4A,B*), indicating that the UNC-13 proteins were expressed at similar levels in these lines. These results suggest that the decreased tonic and evoked release in UNC-13LΔC2A rescued animals are unlikely to result from reduced UNC-13L abundance. Prior studies have shown that the C2A domain is crucial for synaptic localization of UNC-13 (*Melom et al., 2013*). Therefore, it is likely that the decrease in release probability in the UNC-13LΔC2A rescue animals was caused by mis-localization of UNC-13 at the active zone.

We next asked if the decreased release probability in the *unc-10*(K77/79E) mutant was caused by a change in the sub-localization of UNC-13L. To determine this, we compared colocalization of UNC-13L and an active zone marker (ELKS-1) in wild-type controls and in *unc-10*(K77/79E) mutants. UNC-13L and ELKS-1 were strongly colocalized in wild-type (*Figure 4C,D*), and this colocalization was not obviously altered in *unc-10*(K77/79E) mutants (*Figure 4E,F*). These results suggest that disrupting UNC-13L binding to RIM does not prevent synaptic localization of UNC-13L.

## Disrupting heterodimerization does not affect synaptic depression and recovery

Because of the reduced $P_{vr}$ in the *unc-10*(K77/79E) mutant, we asked whether the synaptic depression and recovery would be altered by disruption of C2A/RIM heterodimerization. An integrated line expressing ChIEF, a variant of channelrhodopsin-2 (ChR2) in the cholinergic motor neurons was used to record the EPSCs evoked by a train or a paired light stimulus (*Liu et al., 2009*; *Watanabe et al., 2013*). Both the control animals and the *unc-10*(K77/79E) mutants exhibited synaptic depression following a 1 Hz or 5 Hz train stimulus (*Figure 5A,C*). The depression rate was described by normalizing the amplitudes of all the EPSCs ($EPSC_i$) to the amplitude of the first EPSC ($EPSC_1$). The *unc-10*(K77/79E) mutants displayed a similar depression rate to that of the control animals (*Figure 5B,D*), indicating that RIM binding to the C2A domain is not required for synaptic depression. To assess synaptic recovery, we applied a paired stimulus by changing the intervals between the two stimuli (50 ms, 100 ms, 200 ms, 500 ms, 1 s, 5 s) (*Figure 5E*). The recovery rate was quantified by the ratio of the second EPSC amplitude ($EPSC_2$) to the first EPSC amplitude ($EPSC_1$). As shown in *Figure 5F*, no difference in the recovery rate was observed between the control animals and the *unc-10*(K77/79E)

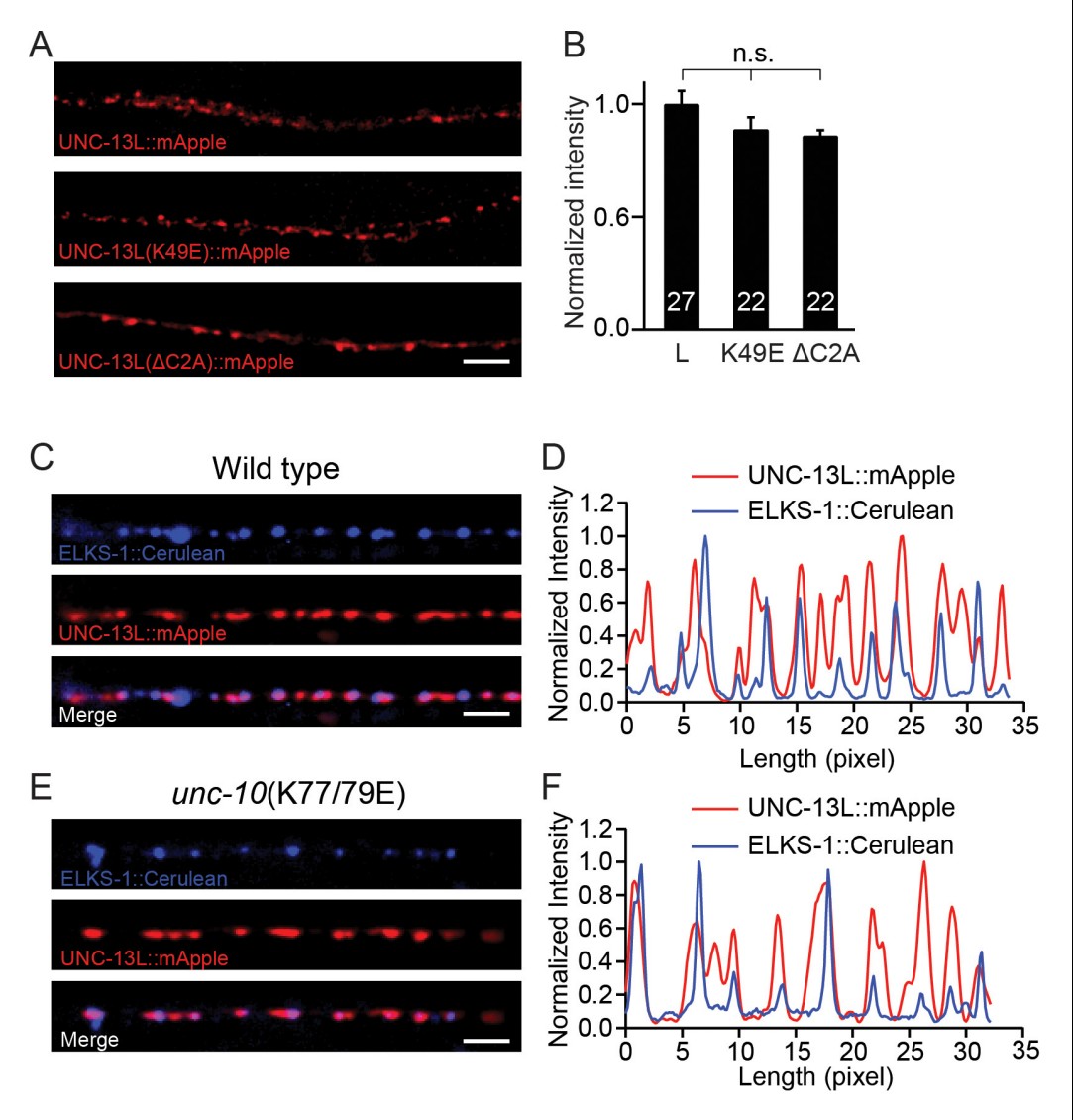

**Figure 4.** Expression of UNC-13 transgenic constructs and effects of disrupting UNC-13L/RIM heterodimerization on sub-localization of UNC-13L. (**A**) Presynaptic distribution of UNC-13 at the dorsal nerve cord in *unc-13* animals rescued by the indicated transgenic constructs. Scale bar, 5 μm. (**B**) Quantification of the mApple fluorescence intensity from the transgenic rescue lines in (**A**). (**C**) Representative confocal Z-stack images for UNC-13L and ELKS-1 in wild-type background. Scale bar, 5 μm. (**D**) Line scans along the dorsal nerve cord. (**E**) Representative confocal Z-stack images for UNC-13L and ELKS-1 in *unc-10*(K77/79E) mutants. (**F**) Line scans along the dorsal nerve cord. Data are mean ± SEM (n.s., non-significant; one-way ANOVA). The number of worms analyzed for each genotype is indicated in the bar graphs.

DOI: https://doi.org/10.7554/eLife.40585.007

The following figure supplement is available for figure 4:

**Figure supplement 1.** UNC-13 proteins with C-terminal tags are fully functional in mediating SV release.
DOI: https://doi.org/10.7554/eLife.40585.008

mutants. This indicates that disrupting C2A/RIM binding does not affect refilling of the SVs at synapses.

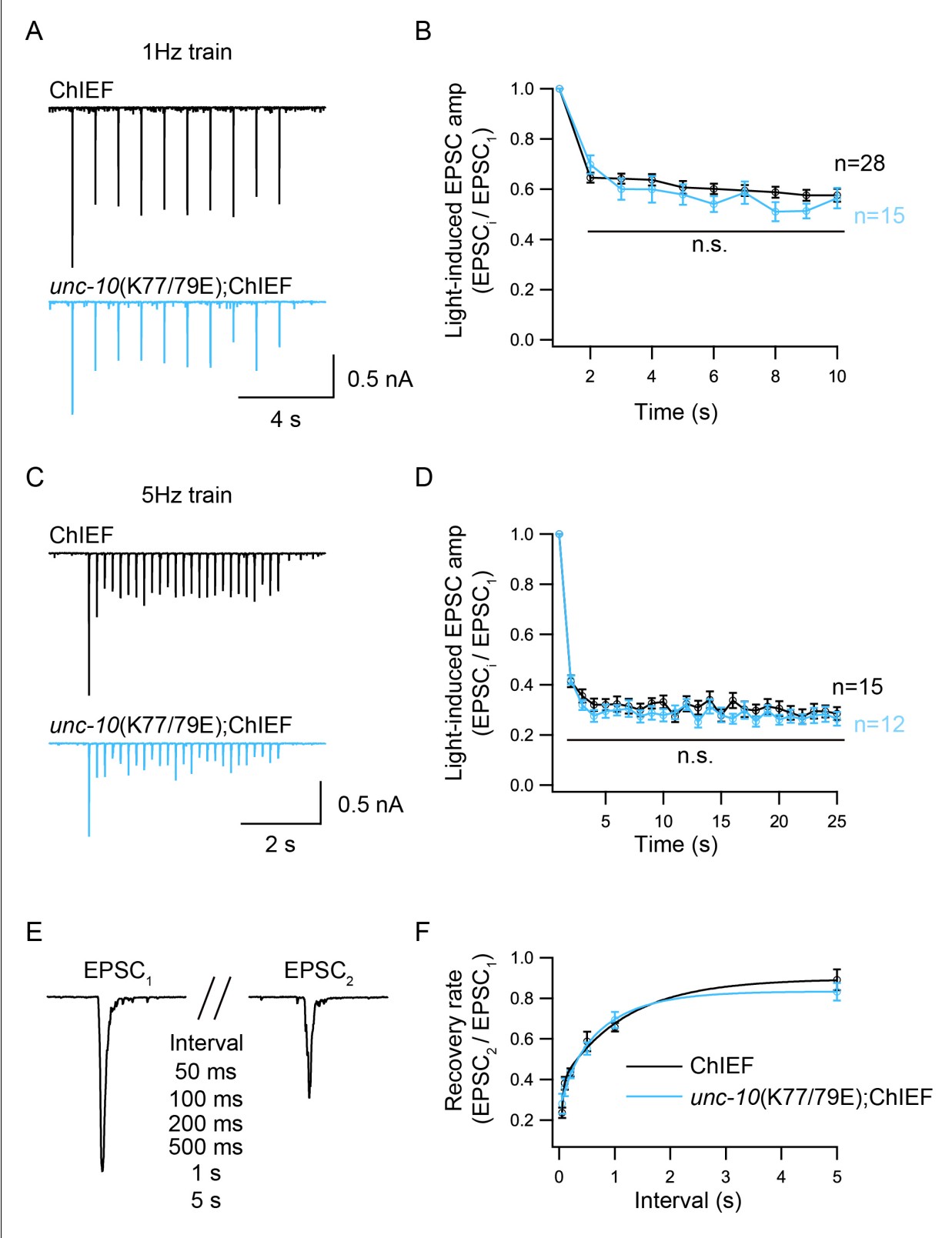

**Figure 5.** Synaptic depression and recovery are unaltered by disrupting C2A/RIM heterodimerization. Synaptic depression and recovery were investigated by applying a train (1 Hz or 5 Hz) or a paired light stimulus onto the ventral nerve cord of the adult worms with expression of ChIEF in cholinergic motor neurons. (A, C) Example traces of 1 Hz and 5 Hz light train stimulus-evoked EPSCs (black, wild-type; red, *unc-10*(K77/79E) mutants). (B, D) Quantification of synaptic depression by normalizing the EPSC amplitude ($EPSC_i$) to the first EPSC amplitude ($EPSC_1$). (E) Evoked EPSCs

*Figure 5 continued on next page*

*Figure 5 continued*

triggered by a paired light stimulus with various intervals ranging from 0.05 s to 5 s. (**D**) Synaptic recovery calculated by the ratio of $EPSC_2$ to $EPSC_1$ (wild-type, 50 ms = 7, 100 ms = 8, 200 ms = 11, 500 ms n = 7, 1 s n = 18, 5 s n = 8; *unc-10*(*nu487* K77/79E) mutant, 50 ms = 5, 100 ms = 7, 200 ms = 11, 500 ms n = 10, 1 s n = 14, 5 s n = 7). Data are mean ± SEM (n.s., non-significant when compared to wild-type; student's t-test).
DOI: https://doi.org/10.7554/eLife.40585.009

## Discussion

In this study, we analyzed the functional importance of the C2A domain of UNC-13L, a Munc13-1 homolog in *C. elegans*, for SV priming and release probability. Prior studies in mouse and *C. elegans* have reported contradictory results regarding the role of the C2A domain in priming (*Betz et al., 2001*; *Zhou et al., 2013a*; *Camacho et al., 2017*). This motivated us to investigate the function of this domain, with a particular focus on its two binding modes, in SV priming and post-priming $Ca^{2+}$-triggered exocytosis. Below we discuss our findings.

### UNC-13L primes SVs independently of the C2A domain

The fact that the UNC-13/Munc13 is required for SV priming has been reported in both invertebrates and vertebrates (*Aravamudan et al., 1999*; *Augustin et al., 1999*; *Richmond et al., 1999*). Sucrose-evoked responses are almost eliminated in neurons lacking UNC-13 or Munc13. Thus far, however, it is still unclear how the priming activity of UNC-13/Munc13 is achieved and regulated. It is believed that the MUN domain plays an essential role in SV priming. Removing the C2A domain from Munc13-1 leads to a marked decrease in the RRP of SVs in cultured hippocampal neurons, suggesting a role for C2A in priming (*Camacho et al., 2017*). However, we found that *unc-13* mutants lacking a C2A domain (UNC-13LΔC2A rescued animals) exhibit a normal RRP, arguing that the C2A domain is not required for priming (*Figure 3E*). Zhou et al. showed that an isolated *unc-13* mutant (*n2609*) with a point mutation in the C2A domain of UNC-13L exhibits a similar reduction in evoked EPSCs to UNC-13LΔC2A, whereas docked SV numbers are normal in the *unc-13(n2609)* mutant. This suggests that the C2A domain is not required for docking (*Zhou et al., 2013a*), consistent with our observations in relation to priming. It should be noted that all the UNC-13 transgenes in this study are extrachromosomal arrays, and that the UNC-13 proteins are overexpressed in these transgenic animals. The overexpression of UNC-13 may cause differential effects with wild-type expression of this protein. Despite this, our findings (e.g., ΔC2A) are similar to the results of Zhou et al., which were obtained from *unc-13* mutants rescued by a single copy of UNC-13.

Our results imply that UNC-13L regulates SV priming via a different mechanism, and that this mechanism is independent of the C2A domain. Prior studies have shown that a point mutation in the C1 domain of Munc13-1 accelerates hypertonicity-induced SV release without changing the size of the RRP (*Basu et al., 2007*). Transgenic *C. elegans* rescued by a truncated UNC-13L lacking its C1 or C2B domain also exhibit a normal RRP (*Michelassi et al., 2017*). Expression of a mutated Munc13-1 (P827L, located in the linker domain between C2B and MUN) leads to a significant increase in release probability without causing an increase in the size of the RRP (*Lipstein et al., 2017*). These observations appear to support the notion that the regulatory domains in UNC-13L (e.g. C2A, C1 and C2B) alter release probability but are not required for SV priming. Thus, our results, together with the findings from other studies in *C. elegans* and mouse, suggest that UNC-13 has multiple regulatory effects on synaptic strength and plasticity, which are achieved by differential control of release probability by its various functional domains.

### State of the C2A domain and release probability

The reduced evoked EPSCs and unchanged RRP in the UNC-13LΔC2A rescue animals led to a significant decrease in release probability ($P_{vr}$), consistent with previous studies (*Camacho et al., 2017*). Similar to these observations, the *unc-10*(K77/79E) mutant also exhibits an impairment in evoked EPSCs but a normal RRP, thereby leading to a decreased $P_{vr}$ (*Figure 3*). These results demonstrate that deleting the C2A domain or blocking C2A/RIM heterodimerization decreases release probability. It should be noted that, release probability remains unchanged in the hippocampal neurons when C2A/RIM heterodimerization is disrupted, because of a similar decrease in the evoked EPSCs and the RRP (*Camacho et al., 2017*), different from our findings in this study.

The function of the C2A domain in regulating the release probability appears to be determined by its state. It has been proposed that the Munc13 C2A domain has three binding states, homo-dimer, heterodimer, and monomer, and that these three forms may co-exist in synaptic terminals. The decrease in evoked EPSC and release probability in the *unc-10*(*nu487* K77/79E) mutant was blocked by disrupting C2A/C2A homodimerization (*Figure 3*). This indicates that the synaptic trans-mission defect caused by disrupting C2A/RIM heterodimerization occurs because of an increase in C2A/C2A homodimerization, which is proposed to comprise an autoinhibitory state. Again, this is post-priming regulation, as SV priming is normal in both the UNC-13L(K49E) rescue and UNC-13L (K49E); *unc-10*(*nu487* K77/79E) animals (*Figure 3E*). Although our findings are similar to the observa-tions reported in cultured hippocampal neurons in which the reduced evoked EPSCs caused by the disruption of heterodimerization were partially recovered when homodimerization and heterodimeri-zation were disrupted simultaneously (*Camacho et al., 2017*), we found that UNC-13L with a mono-meric C2A domain is fully functional for both the priming and post-priming $Ca^{2+}$-triggered tonic and evoked release. Thus, our results support the notion that homodimerization of the C2A domain inhibits SV release, and that RIM releases the C2A domain from the autoinhibitory homodimeric complex by competitively binding the C2A domain to form a heterodimer, thereby allowing the C2A domain to become fusion-competent. The heterodimer itself, however, does not produce an addi-tional effect on synaptic transmission.

## Distinct priming mechanisms between worm and mouse?

RIM binding to Munc13-1 is a key determinant in SV priming in the mouse central nervous system. In fact, deleting Munc13-1 C2A or disrupting C2A/RIM heterodimerization leads to a similar reduction in the size of the RRP to that observed in RIM knockout neurons (*Deng et al., 2011*; *Camacho et al., 2017*). Whereas SV priming in worm does not rely on the UNC-13 C2A and the heterodimerization of UNC-13/RIM, although this interaction is highly conserved between worm and mouse. This raises a question of whether the UNC-10/RIM is required for priming in worm. Electron microscopy results have revealed that the number of docked SVs is slightly decreased in the *unc-10* mutant (*Koushika et al., 2001*; *Weimer et al., 2006*; *Gracheva et al., 2008*), which cannot account for the far more dramatic SV fusion defect, suggesting a post-docking role of RIM in the active zone. Con-sistent with these relatively minor docking defects, we found that the size of the RRP in *unc-10* mutants (assessed by sucrose evoked currents) is comparable to that in the wild-type animals (data not shown), demonstrating that UNC-10/RIM is not required for SV priming. These results suggest that UNC-10/RIM primarily regulates SV fusion by increasing release probability and not by promot-ing SV priming. UNC-10/RIM increases release probability via its ability to bind UNC-13 but also through its requirement for recruiting CaV2 calcium channels to presynaptic terminals (*Han et al., 2011*; *Kaeser et al., 2011*; *Müller et al., 2012*).

Why does RIM play different roles at worm and mouse synapses? SV priming is thought to consist of two sequential steps, termed molecular and positional priming (*Neher and Sakaba, 2008*). Molecular priming is proposed to account for a slow component of release, which consists of fusion of SVs that are docked far from calcium channels driving release. Positional priming is proposed to mediate a faster form of release, which comprises fusion of SVs that are docked very close to calcium channels. One possibility is that worms rely less on positional priming, which presumably requires UNC-13/RIM interaction, and instead rely more on molecular priming. In this scenario, worm *unc-10* RIM mutants would have decreased release probability but relatively normal RRP size. By contrast, mice may require positional priming to maintain the RRP, and thus RIM mutants would have signifi-cant priming defects.

## Additional functions of the C2A domain

Despite the above findings, the actual mechanism whereby the UNC-13L with a monomeric C2A regulates SV release remains unclear. Based on the normal synaptic transmission in the UNC-13L (K49E); *unc-10*(*nu487* K77/79E) double mutants, RIM binding to the UNC-13L C2A domain only appears to be playing a switch role. The C2A domain must therefore have additional functions for regulation of SV release. This is confirmed by our observations in relation to tonic release, which has been proposed to have a different regulatory mechanism from evoked release (*Ramirez and Kava-lali, 2011*; *Melom et al., 2013*). The mEPSC frequency was reduced by 50% in the UNC-13LΔC2A

rescue animals at both 0 mM and 1 mM $Ca^{2+}$, similar to the evoked release findings (*Figure 2*), revealing an important role of the C2A domain for tonic release at cholinergic synapses. However, this role cannot be explained by the state of the C2A domain (homodimer, heterodimer, or monomer), as the mEPSC frequencies were normal in the UNC-13L(K49E) rescue, *unc-10*(*nu487* K77/79E) mutant, and UNC-13L(K49E); *unc-10*(*nu487* K77/79E) double mutants in 1 mM $Ca^{2+}$. We only observed a change in the $Ca^{2+}$ sensitivity in very low $Ca^{2+}$ at GABAergic synapses, and this phenotype disappeared when the $Ca^{2+}$ levels were increased (*Figure 2G*). Our observations at cholinergic synapses therefore demonstrate that, although the C2A/C2A homodimer inhibits evoked release, it does not affect tonic release. There must still be an unknown mechanism by which the monomeric C2A domain regulates post-priming SV exocytosis.

## Materials and methods

### Strains

Strain maintenance and genetic manipulation were performed as previously described (*Brenner, 1974*). Animals were cultivated at room temperature on nematode growth medium (NGM) agar plates seeded with OP50 bacteria. On the day before experiments L4 larval stage animals were transferred to fresh plates seeded with OP50 bacteria for all the electrophysiological recordings. The following strains were used:

Wild-type, N2 bristol
KP6901 *unc-13(s69)* I
KP7503 *nu487 unc-10*(*nu487* K77/79E), X
KP6893 nuEx1515 [P*snb-1*::UNC-13L];*unc-13(s69)*
ZTH6 *unc-13(s69)*;hztEx106 [P*snb-1*::UNC-13LΔC2A]
ZTH454 *unc-13(s69)*;hztEx91 [P*snb-1*::UNC-13L(K49E)]
ZTH7 *unc-10*(*nu487* K77/79E);hztEx91 [P*snb-1*::UNC-13L(K49E); *unc-13(s69)*]
ZX460 zxIs6 [P*unc-17*::ChR2(H134R)::YFP +*lin-15*(+)] V
ZTH71 *unc-10*(*nu487* K77/79E);zxIs6
ZTH605 *unc-13(s69)*;hztEx76 [P*snb-1*::UNC-13L::mApple]
ZTH606 *unc-13(s69)*;hztEx77 [P*snb-1*::UNC-13LΔC2A::mApple]
ZTH607 *unc-13(s69)*;hztEx78 [P*snb-1*::UNC-13L(K49E)::mApple]
JSD0025 tauIs12 (9x) [P*unc-129*::ELKS-1::Cerulean::Cerulean LG]
EG5793 oxSi91[P*unc-17*::ChIEF::mCherry::*unc-54*UTR; *unc-119*(+)] II
ZTH608 *unc-10*(*nu487* K77/79E);oxSi91, MosSCI
'Ex' in the strain name indicates that the rescue experiments use extrachromosomal array.

### Constructs, transgenes and germline transformation

A 5.451 kb cDNA corresponding to UNC-13L was amplified by PCR from the N2 cDNA library and inserted into MCSII of the JB6 vector between the KpnI and NotI sites. The *snb-1* promoter (3 kb) was inserted into MCSI between the SphI and BamHI sites. The C2A domain deletion removed the first 150 amino acids. The 5' primer (GCTGCAGACTGTAGAATCAACAACTGT) and 3' primer (ACAGTTGTTGATTCTACAGTCTGCAGC) were used to create the P*snb-1*::UNC-13L(K49E) construct.

Transgenic strains were isolated by microinjection of various plasmids using either Pmyo-2::NLS-GFP (KP#1106) or Pmyo-2::NLS-mCherry (KP#1480) as the co-injection marker. Integrated transgenes were obtained by UV irradiation of strains carrying extrachromosomal arrays. All integrated transgenes were outcrossed at least seven times.

### Electrophysiology

Electrophysiology was conducted on dissected *C. elegans* as previously described (*Hu et al., 2012*). Worms were superfused in an extracellular solution containing 127 mM NaCl, 5 mM KCl, 26 mM $NaHCO_3$, 1.25 mM $NaH_2PO_4$, 20 mM glucose, 1 mM $CaCl_2$, and 4 mM $MgCl_2$, bubbled with 5% $CO_2$, 95% $O_2$ at 22°C. The 1 mM $CaCl_2$ was replaced by 1 mM $MgCl_2$ to record mEPSCs and mIPSCs in 0 mM of $Ca^{2+}$. Whole-cell recordings were carried out at −60 mV for all EPSCs, including mEPSCs, evoked EPSCs, and sucrose-evoked responses. The holding potential was switched to 0 mV to record mIPSCs. The internal solution contained 105 mM $CH_3O_3SCs$, 10 mM CsCl, 15 mM CsF, 4 mM

MgCl$_2$, 5 mM EGTA, 0.25 mM CaCl$_2$, 10 mM HEPES, and 4 mM Na$_2$ATP, adjusted to pH 7.2 using CsOH. Stimulus-evoked EPSCs were stimulated by placing a borosilicate pipette (5–10 µm) near the ventral nerve cord (one muscle distance from the recording pipette) and applying a 0.4 ms, 85 µA square pulse using a stimulus current generator (WPI). A pulsed application of sucrose (1M, 2 s) was applied onto the ventral nerve cord using a Picospritzer III (Parker). The glass pipette containing sucrose was placed at the end of the patched muscle (around half muscle distance from the recording pipette).

## Fluorescence imaging

Animals were immobilized on 2% agarose pads with 30 mM levamisole. Fluorescence imaging was performed on a spinning-disk confocal system (3i Yokogawa W1 SDC) controlled by Slidebook 6.0 software. Animals were imaged with an Olympus 100 × 1.4 NA Plan-Apochromat objective. Z series of optical sections were acquired at 0.11 µm steps. Images were deconvolved with Huygens Professional version 16.10 (Scientific Volume Imaging, The Netherlands) and then processed to yield maximum intensity projections using ImageJ 1.51 n (Wayne Rasband, National Institutes of Health).

## Isothermal titration calorimetry (ITC)

ITC experiments were performed using a Nano ITC (TA Instruments). ITC titrations were performed in degassed 20 mM HEPES pH 7.5, 150 mM NaCl at 25°C. After baseline subtraction, the area under each peak was integrated and data were fit to a one site independent binding model usingNanoAnalyze software (TA Instruments) to determine the binding affinity (K), enthalpy (dH) and stoichiometry (n); binding was not observed for unc-10 62–121 K77/79E.

 Protein expression and purification (for ITC experiments): C. elegans UNC-13 long isoform residues 1–258 bearing the K49E mutation (which abolishes homodimer formation), C. elegans UNC-10 residues 62–121 WT or C. elegans UNC-10 K77/79E were expressed from pGEX6p1 as a GST fusion in Rosetta2 (DE3) cells. Cells were grown at 37°C in 6L 2xyt with carbenicillin (100 µg/mL), chloramphenicol (35 µg/mL) and ZnSO$_4$ (200 µM, Unc-10 constructs only), to O.D.600 ~ 0.5, then allowed to cool down to 18°C for an hour before induction with 1 mM IPTG overnight at 18°C. The cells were harvested the next day and lysed in 20 mM HEPES (pH 7.5), 150 mM NaCl, 1 mM PMSF, 1% Triton X-100, 1 mM DTT, 200 µM ZnSO$_4$ (UNC-10 constructs only); the cells were sonicated 3 × 40 s using a midi tip, and lysates clarified 45 mins @ 40,000 rpm, 4°C in a Sorvall Type 45 Ti rotor. Clarified lysates were bound to 1.5 mL (bed volume) glutathione sepharose beads tumbling overnight at 4°C. The next day, the beads were washed extensively, and HRV3C protease added for on-beads cleavage overnight at 4°C. The eluted soluble constructs were concentrated and subjected to gel filtration on a Superdex 75 10/300 against 20 mM HEPES pH 7.5 150 mM NaCl at 4°C. Single peaks corresponding to the monomeric proteins were concentrated to 21 µM (UNC-13L residues 1–258), to 140 µM (UNC-10 62–121 WT) and to 116 µM (UNC-10 62–121 K77/79E mutant), and used in the ITC experiment.

## Data acquisition and statistical analysis

All recordings were obtained using a HEKA EPC10 double amplifier (HEKA Elektronik) filtered at 2 kHz, and analyzed by built-in programs in Igor seven software (WaveMetrics). Each set of data represents the mean ± SEM of an indicated number (n) of animals. Statistical significance was determined using student's-t test or one-way ANOVA followed by Dunnett's test to control for multiple comparisons.

## Acknowledgements

We thank the C. elegans Genetics Stock Center for strains and reagents. We thank members of the Hu lab. We thank Rowan Tweedale for critically reading the manuscript. This work was supported by an Australia Research Council Discovery Project grant (DP160100849 to ZH), a National Health and Medical Research Council Project grant (APP1122351 to ZH), a NARSAD Young Investigator grant (24980 to ZH), a National Institutes of Health research grant (GM54728 to JK), an Autism Science Foundation (REG15-006 to DN) and a William Randolph Hearst Postdoctoral Fellowship (to DN).

## Additional information

### Funding

| Funder | Grant reference number | Author |
|---|---|---|
| Australian Research Council | DP160100849 | Zhitao Hu |
| National Health and Medical Research Council | APP1122351 | Zhitao Hu |
| Brain and Behavior Research Foundation | 24980 | Zhitao Hu |
| National Institutes of Health | GM54728 | Joshua M Kaplan |
| Austism Science Foundation | REG15-006 | Daniel Nedelcu |
| William Randolph Hearst Post-doctoral Fellowship | | Daniel Nedelcu |

The funders had no role in study design, data collection and interpretation, or the decision to submit the work for publication.

### Author contributions

Haowen Liu, Lei Li, Conception and design, Acquisition of data, Analysis and interpretation of data; Daniel Nedelcu, Qi Hall, Lijun Zhou, Acquisition of data, Analysis and interpretation of data; Wei Wang, Yi Yu, Conception and design, Contributed unpublished essential data or reagents; Joshua M Kaplan, Zhitao Hu, Supervision, Conception and design, Analysis and interpretation of data, Drafting or revising the article

### Author ORCIDs

Lijun Zhou (iD) http://orcid.org/0000-0002-0393-4787
Joshua M Kaplan (iD) http://orcid.org/0000-0001-7418-7179
Zhitao Hu (iD) http://orcid.org/0000-0002-2948-3339

### Decision letter and Author response

Decision letter https://doi.org/10.7554/eLife.40585.012
Author response https://doi.org/10.7554/eLife.40585.013

## Additional files

### Supplementary files

• Transparent reporting form
DOI: https://doi.org/10.7554/eLife.40585.010

### Data availability

All data generated or analysed during this study are included in the manuscript and supporting files.

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
