## [Decision Letter]

Thank you for submitting your article "Heterodimerization of UNC-13/RIM regulates synaptic vesicle release probability but not priming" for consideration by *eLife*. Your article has been reviewed by three peer reviewers, including Josep Rizo as the Reviewing Editor, and the evaluation has been overseen by Randy Schekman as the Senior Editor. The following individuals involved in the review of your submission have agreed to reveal their identity: Nils Brose (Reviewer #2) and Janet Richmond (reviewer #3).

The reviewers have discussed the reviews with one another and the Reviewing Editor has drafted this decision to help you prepare a revised submission.

Summary:

Liu et al. present a very interesting story on a very important aspect of synapse biology: the mechanism by which the interplay between UNC-13 and UNC-10 determines presynaptic function. UNC-13/Munc13 proteins are essential regulators of synaptic vesicle priming. Their loss causes a complete arrest of transmitter release, and their regulation by protein-interactors or second messengers determines several key aspects of synapse function, most notably short-term synaptic plasticity. In this regard the focus on the UNC-13/UNC-10 interaction is important and of very substantial interest because the accounts in the literature are confusing, with evidence for (and against) roles in synapse targeting and sub-synaptic localization of UNC-13/Munc13 and in the regulation of UNC-13/Munc13 dimerization and UNC-13/Munc13 activity. Liu et al. show that the UNC-13L C2A domain homodimerizes and also heterodimerizes with UNC-10, in correlation with previous results obtained with the mammalian proteins. Their data suggest that the UNC-13L C2A domain and its interaction with UNC-10 are important to increase the release probability but do not control priming. The latter result contrasts with data described in Camacho et al., 2017. The reasons for this apparent discrepancy are not clear, but the reviewers believe that the present results will be of substantial interest to the field of synapse biology and, overall, are supportive of the manuscript. However, they also have a number of concerns that are summarized below.

Essential revisions:

1) Earlier studies showed that a truncation of mammalian Munc13-1 that abolishes RIM1 binding leads to a priming defect that can be (partially) overcome by prolonged strong overexpression of the truncated Munc13-1 variant (Betz et al., – 2001 –). This indicated that the Munc13/RIM interaction might play a role in the synaptic trafficking and sub-synaptic positioning and anchoring of Munc13s. This notion is supported by the fact that Munc13-1 and ubMunc13-2 levels are massively reduced in RIM1 KO mice. Subsequent analyses in RIM1 KOs showed that synaptic targeting and active zone anchoring of Munc13-1 (and of the other RIM-binding Munc13, ubMunc13-2) are perturbed in the absence of RIM1 (Andrews-Zwilling et al. – 2006 – J. Biol. Chem. 281, 19720). In a parallel study, it was shown that perturbation of the Munc13-RIM interaction by infusion of the isolated RIM2 ZnF domain strongly inhibits transmitter release from the calyx of Held while a Munc13-binding-deficient ZnF domain variant does not (Dulubova et al., – 2005 –), which corroborates the earlier/parallel work cited above. Since the first study on the functional effects of the RIM-Munc13 interaction on Munc13-dimerization (Deng et al., – 2011 –), the whole field focused almost exclusively on this issue, without addressing the effects of perturbing RIM/Munc13 or UNC-13/UNC-10 interactions on Munc13/UNC-13 localization (beyond the analysis of the mere presence or absence of Munc13/UNC-13 at synapses). This is unfortunate because we know that the levels of Munc13/UNC-13 at synapses and its sub-synaptic positioning affect key synapse features. In addition, previous studies in *C. elegans* showed altered distributions of docked vesicles away from the active zone in the presence of UNC-13 constructs lacking the C2A domain as well as UNC-10 mutants, and this positional effect likely accounts for the reduced release probability observed in the present study. Liu et al. did not analyze sub-cellular localization and did not present expression data to determine the levels of the protein constructs reintroduced in the nerve cord. This is problematic given that all of the UNC-13 constructs are extrachromasomal arrays and thus could be highly or differentially expressed. Furthermore, the present study would have benefited from using CRISPR or some other form of homologous recombination throughout, as was done for their UNC-10 (K77/79E) construct so that expression level effects do not confound the interpretation of the results. Of particular concern is the possibility that overexpression of UNC-13 K49E in the rescue of release in the UNC-10 (K77/79E) background may not be physiologically relevant.

For these reasons, the present study and its interpretation would profit tremendously from a comparative analysis of the expression levels and sub-cellular localization of the different UNC-13 variants at synapses (as well as of UNC-13 levels and positioning in the presence of the different UNC-10 variants). If analyzing sub-synaptic UNC-13 localization cannot be performed within a reasonable timeframe for this communication, the authors should at least quantify the synaptic UNC-13 levels. The authors should also discuss these various issues and tone down their conclusions accordingly (including the definitive nature of the title).

2) Crucial differences observed between the results described by Liu et al. and by Camacho et al., 2017 correspond to the measurements of the readily-releasable pool (RRP) of vesicles by treatment with hypertonic sucrose. It is unclear whether these are indeed real differences between the mammalian and worm systems, or the different results arise from differences in experimental protocols. For instance, the authors use 1 M sucrose instead of 500 mM sucrose, which is more common and was used by Camacho et al. It is also plausible that differences might arise from the method of application of sucrose. The authors should perform experiments with 500 mM sucrose to examine whether their results stand at this lower concentration, provide more experimental details on these experiments and discuss how experimental details might influence the results obtained. Raw traces for both concentrations should be provided and, when collecting the 500 mOsm data, the leak current for each trace used in the analysis should be noted.

[Editors' note: further revisions were requested prior to acceptance, as described below.]

Thank you for submitting your article "Heterodimerization of UNC-13/RIM regulates synaptic vesicle release probability but not priming in *C. elegans*" for consideration by *eLife*. Your article has been reviewed by two peer reviewers, and the evaluation has been overseen by a Reviewing Editor and Randy Schekman as the Senior Editor. The following individuals involved in review of your submission have agreed to reveal their identity: Nils Brose (Reviewer #2); Janet E Richmond (Reviewer #3).

The reviewers have discussed the reviews with one another and with the Reviewing Editor. They found most revisions satisfactory, but there are still some concerns that are summarized below to help you prepare a revised submission.

1) The authors produced new extrachromasomal lines for the UNC-13 constructs tagged with C-terminal Apple, which allowed them to measure comparable expression levels, but these were not the arrays used to do the functional electrophysiological analysis in the paper. It is the case that arrays can vary in expression levels from one injection to the next. The authors only provide assurances that the behavior was rescued but did not show the data or the extent of the rescue. More concerning, they did not repeat the electrophysiological analyses on these strains. Without this, how can we be certain that these are similar to the original array-expressing strains and that the same results would be found.

2) As for the hyperosmotic traces, only those for the unc-10(k77/79E) were reexamined in 0.5M sucrose and the requested leak currents for the two data sets (i.e. 0.5 and 1M) were not provided. The concern is that at 1M sucrose, if there were substantial leak, the large inward currents observed would arise not from sucrose-induced release but poor seals. This problem has arisen in the past in analyses of fly embryos in which large inward currents were observed in mutants that were subsequently shown to have no primed vesicles; the reason was leaky recordings. If the authors could have produced these data for the *unc-13* null and the UNC-13 rescuing arrays tagged with Apple, this would have greatly improved confidence in the conclusions of the paper. The leak currents should be provided and the data should be interpreted considering these issues.

3) In the decision letter a critical discussion of the use of extrachromosomal-array-based expression was provided. Clearly, overexpression via this approach could mask effects that would become detectable with WT-like expression levels. This issue should be discussed briefly as a possible caveat.

---

## [Author Response]

Essential revisions:1) Earlier studies showed that a truncation of mammalian Munc13-1 that abolishes RIM1 binding leads to a priming defect that can be (partially) overcome by prolonged strong overexpression of the truncated Munc13-1 variant (Betz et al., 2001).[…] Furthermore, the present study would have benefited from using CRISPR or some other form of homologous recombination throughout, as was done for their UNC-10 (K77/79E) construct so that expression level effects do not confound the interpretation of the results. Of particular concern is the possibility that overexpression of UNC-13 K49E in the rescue of release in the UNC-10(K77/79E) background may not be physiologically relevant.For these reasons, the present study and its interpretation would profit tremendously from a comparative analysis of the expression levels and sub-cellular localization of the different UNC-13 variants at synapses (as well as of UNC-13 levels and positioning in the presence of the different UNC-10 variants). If analyzing sub-synaptic UNC-13 localization cannot be performed within a reasonable timeframe for this communication, the authors should at least quantify the synaptic UNC-13 levels. The authors should also discuss these various issues and tone down their conclusions accordingly (including the definitive nature of the title).

We agree that the synaptic levels and localization of Munc13/UNC-13 proteins at the nerve terminals are essential for SV exocytosis. The use of extrachromosomal arrays in this study raised the question of whether the UNC-13 protein levels from distinct transgenic lines (UNC-13L, UNC-13LK49E, and UNC-13LDC2A rescues) are similar or not.

To determine this, we made all the rescue constructs with a fluorescence protein tag (mApple). Re-introducing these constructs into the *unc-13* null mutants rescued the locomotion defects (data not shown). Indeed, the UNC-13 with a C-terminal tag is fully functional in mediating SV exocytosis (Hu et al., 2013). By analyzing the fluorescence intensity of mApple, we showed that the average proteins levels of UNC-13L, UNC13LK49E, and UNC-13LDC2A are not significantly different (Figure 4A, B). These results exclude the possibility that the decreased SV release in the UNC-13LDC2A rescue is due to a change in the protein level, and that disrupting the C2A/C2A homodimerization has no effect on protein level. It should be noted that we didn’t compare release in these new mApple transgenic strains, as prior studies have shown that a fluorescence tag does not affect UNC-13 protein’s function (Madison et al., 2005; Hu et al., 2013).

To examine whether the sub-localization of UNC-13L is altered when heterodimerization with RIM is disrupted, we analyzed UNC-13L co-localization with ELKS1, another active zone protein. In wild-type background, UNC-13L (tagged with mCherry) and ELKS^-1^ (tagged with Cerulean) exhibit an obvious colocalization (Figure 4C, D). This colocalization was still observed in *unc-10*(K77/79E) mutants (Figure 4E, F). These results suggest that disrupting the UNC-13L/RIM interaction does not dramatically alter UNC-13L synaptic localization and that UNC-13L likely has other binding partners that promote its active zone localization.

All the above results have been incorporated into the figures and text.

“The decreased release probability in the UNC-13LDC2A rescue and the *unc-10*(K77/79E) mutant could arise from a change of the expression level of UNC-13, or a change of the sublocalization of the UNC-13 proteins at the nerve terminals. […] These results suggest that disrupting UNC-13L binding to RIM does not prevent synaptic localization of UNC-13L.”

The new generated imaging lines have been added into the “Strain names” in the Materials and methods section.

2) Crucial differences observed between the results described by Liu et al. and by Camacho et al., 2017 correspond to the measurements of the readily-releasable pool (RRP) of vesicles by treatment with hypertonic sucrose. It is unclear whether these are indeed real differences between the mammalian and worm systems, or the different results arise from differences in experimental protocols. For instance, the authors use 1 M sucrose instead of 500 mM sucrose, which is more common and was used by Camacho et al. It is also plausible that differences might arise from the method of application of sucrose. The authors should perform experiments with 500 mM sucrose to examine whether their results stand at this lower concentration, provide more experimental details on these experiments and discuss how experimental details might influence the results obtained. Raw traces for both concentrations should be provided and, when collecting the 500 mOsm data, the leak current for each trace used in the analysis should be noted.

Since Rosemund and colleagues first established the assay of using hypertonic sucrose-evoked synaptic current to estimate the size of the readily releasable vesicular pool (RRP), this assay has been widely used in both vertebrates and invertebrates. The sucrose concentrations and puffing durations, however, vary in different studies (Aravamudan et al., 1999; Richmond et al., 1999; Yoshihara and Littleton, 2002; Pang et al., 2006; Zhou et al., 2013). It is unlikely that the different results in SV priming in the *unc-10*(K77/79E) arise from the experimental protocols we used, as the large priming defect in the *unc-13* mutants has been consistently observed in our lab. Nevertheless, to address the reviewer’s question, we compared the synaptic currents elicited by 0.5M sucrose between wild type and the *unc10(K77/79E)* mutants. To record a relatively stable sucrose response under this condition, a 5s puffing duration was used (Author response image 1). As shown in Author response image 1, the charge transfer of the sucrose currents in the *unc-10(K77/79E)* mutants were comparable to those in wild-type worms (Author response image 1), consistent with the observation when using 1M sucrose. We have added these results in the main text. All the raw traces recorded in both 0.5M and 1M sucrose were shown in Author response image 1). These results confirmed the notion that disrupting the UNC-13L/RIM heterodimerization does not affect SV priming.

We have described more details how to perform the sucrose puff experiments at the worm NMJ in the “Materials and methods”.

“A pulsed application of sucrose (1M, 2s) was applied onto the ventral nerve cord using a Picospritzer III (Parker). The glass pipette containing sucrose was placed at the end of the patched muscle (around half muscle distance from the recording pipette).”

“Nevertheless, we assessed the RRP size in the UNC-13L(K49E) rescue animals and the *unc10*(K77/79E) mutant by puffing hypertonic sucrose (1M, 2s). […] The unchanged SV priming in the *unc-10*(K77/79E) mutant was also confirmed by puffing a lower concentration of sucrose (0.5M, 5s; data not shown).”

[Editors' note: further revisions were requested prior to acceptance, as described below.]

The reviewers have discussed the reviews with one another and with the Reviewing Editor. They found most revisions satisfactory, but there are still some concerns that are summarized below to help you prepare a revised submission.1) The authors produced new extrachromasomal lines for the UNC-13 constructs tagged with C-terminal Apple, which allowed them to measure comparable expression levels, but these were not the arrays used to do the functional electrophysiological analysis in the paper. It is the case that arrays can vary in expression levels from one injection to the next. The authors only provide assurances that the behavior was rescued but did not show the data or the extent of the rescue. More concerning, they did not repeat the electrophysiological analyses on these strains. Without this, how can we be certain that these are similar to the original array-expressing strains and that the same results would be found.

Both mEPSCs and evoked EPSCs were analyzed in the *unc-13* mutants rescued by mApple-tagged UNC-13 proteins (UNC-13L, DC2A, and K49E). Basically, we found very similar results in these rescue strains, compared to those observed untagged UNC-13 rescue strains, indicating that the UNC-13 with C-terminal tag is fully functional. These results have been incorporated into the revised manuscript as a supplementary figure (Figure 4—figure supplement 1).

“All mApple-tagged UNC-13L constructs rescued the SV release in the *unc-13* mutants, and the phenotypes were similar to those rescued by corresponding untagged UNC13L constructs (Figure 4—figure supplement 1)”

2) As for the hyperosmotic traces, only those for the unc-10(k77/79E) were reexamined in 0.5M sucrose and the requested leak currents for the two data sets (i.e. 0.5 and 1M) were not provided. The concern is that at 1M sucrose, if there were substantial leak, the large inward currents observed would arise not from sucrose-induced release but poor seals. This problem has arisen in the past in analyses of fly embryos in which large inward currents were observed in mutants that were subsequently shown to have no primed vesicles; the reason was leaky recordings. If the authors could have produced these data for the unc-13 null and the UNC-13 rescuing arrays tagged with Apple, this would have greatly improved confidence in the conclusions of the paper. The leak currents should be provided and the data should be interpreted considering these issues.

We now added the quantification for two more mutants, *unc-13(s69)* and UNC-13L rescue, under the lower sucrose condition (0.5M, 5s). The results are similar with what we obtained when puffing 1M sucrose (see Author response image 2). Moreover, Zhou et al. have shown that removing the C2A domain from UNC-13L does not cause a change in the RRP size measured by puffing 0.5M sucrose (Zhou et al., 2013, *eLife*), consistent with our findings in this study. We hope that the reviews can agree that our current protocol for sucrose puffing (1M, 2s) is reasonable.

**Author response image 2. respfig2:** 

We remade the following figures with all sucrose traces. The leak current can be seen on each individual traces. We also quantified the leak current for all the sucrose recordings (1M, 2s). The results have been incorporated into the revised manuscript as a supplementary figure (Figure 3—figure supplement 1). As you can see, the leak current is controlled very well in our recordings. The averaged leak current for each genotype is between -20pA and -30pA.

“It should be noted that recording of the sucrose-evoked current requires a tight gigaohm seal between the recording pipette and the cell, as a loose seal often leads to an unreal current. We therefore analyzed the leak currents for each sucrose recordings. Our data showed that all recordings were controlled well with acceptable leak currents (Figure 3—figure supplement 1).”

3) In the decision letter a critical discussion of the use of extrachromosomal-array-based expression was provided. Clearly, overexpression via this approach could mask effects that would become detectable with WT-like expression levels. This issue should be discussed briefly as a possible caveat.

We agree with this point and have discussed the issue of rescue using extrachromosomal arrays.

“It should be noted that all the UNC-13 transgenes in this study are extrachromosomal arrays, and that the UNC-13 proteins are overexpressed in these transgenic animals. The overexpression of UNC-13 may cause differential effects with wild-type expression of this protein. Despite this, our findings (e.g., DC2A) are similar with Zhou et al. results which were obtained from *unc-13* mutants rescued by single copy of UNC-13.”

**Author response image 3. respfig3:** 

**Author response image 4. respfig4:** 

**Author response image 5. respfig5:** 

**Author response image 6. respfig6:**